# Semi-Supervised Detection, Identification and Segmentation for Abdominal Organs

Mingze Sun, Yankai Jiang, and Heng Guo

Alibaba DAMO Academy
`gh205191@alibaba-inc.com`

**Abstract.** Abdominal organ segmentation is an important prerequisite in many medical image analysis applications. Methods based on U-Net have demonstrated their scalability and achieved great success in different organ segmentation tasks. However, the limited number of data and labels hinders the training process of these methods. Moreover, traditional U-Net models based on convolutional neural networks suffer from limited receptive fields. Lacking the ability to model long-term dependencies from a global perspective, these methods are prone to produce false positive predictions. In this paper, we propose a new semi-supervised learning algorithm based on the vision transformer to overcome these challenges. The overall architecture of our method consists of three stages. In the first stage, we tackle the abdomen region location problem via a lightweight segmentation network. In the second stage, we adopt a vision transformer model equipped with a semi-supervised learning strategy to detect different abdominal organs. In the final stage, we attach multiple organ-specific segmentation networks to automatically segment organs from their bounding boxes. We evaluate our method on MICCAI FLARE 2022 challenge dataset. Experimental results demonstrate the effectiveness of our method. Our segmentation results currently achieve 0.897 mean DSC on the leaderboard of FLARE 2022 validation set.

**Keywords:** Semi-supervised · Organ detection · Organ segmentation.

## 1 Introduction

Learning feature representations from a few labeled data is a fundamental problem in medical image analysis. It has attracted the interest of academia and industry because acquiring enough annotated medical images is tedious, time-consuming, and expensive. Compared to supervised methods, semi-supervised methods mainly focus on using labeled and large amounts of unlabeled data efficiently and properly [3,5,14]. Nowadays, semi-supervised methods are becoming the standard choice for data label shortage regimes.

Deep learning has been very popular in the field of medical image analysis. Modern deep learning-based strong baselines for medical image analysis are mostly trained on a large amount of manually labeled data and tailored for

specific tasks. Abdominal organ segmentation is one of the most common tasks in this subject, which has many important clinical applications, such as organ quantification, surgical planning, and disease diagnosis. However, the shortage of labeled data hinders the development of deep learning models in this scenario since segmentation tasks often require enough dense annotations which come from domain experts' concentration and are hard to access. In addition, the diversity of data sources also challenges the robustness of existing state-of-the-art (SOTA) methods. As a potential alternative, semi-supervised learning can explore useful information from unlabeled cases. Therefore, exploiting unlabeled medical data in a semi-supervised learning scheme has become extremely important to improve the performance of medical image segmentation models and has attracted increasing research attention.

In this paper, we propose a new semi-supervised learning algorithm based on the vision transformer to overcome the aforementioned challenges. The architecture of our method consists of three stages. In the first stage, we build a lightweight segmentation network to locate the abdomen region. Then, in the second stage, we adopt a vision transformer model equipped with a semi-supervised learning strategy to detect different abdominal organs. In the third stage, we attach multiple organ-specific segmentation networks to automatically segment organs from their bounding boxes. We evaluate our method on MICCAI FLARE 2022 challenge dataset. Experimental results demonstrate the effectiveness of each network component in our method. The contributions of our method are threefold: (1) We propose a semi-supervised learning scheme that adopts multiple models' consistent predictions to produce high-quality pseudo labels to train the student network. (2) We propose a vision transformer-based detection model to detect different organs which has large variations in shape and texture. (3) Combining a semi-supervised training strategy and a vision transformer architecture with several segmentation heads, we build a strong segmentation inference framework which currently achieves 0.897 mean DSC on the leaderboard of FLARE 2022 validation set.

## 2    Method

In order to leverage the unlabeled data, we first train a teacher model using labeled data and then predict segmentation results for unlabeled data with the trained teacher model. Considering that many new network structures may not have good generalization ability in the unseen dataset, we choose a strong and general baseline, nnU-Net [7], as the standard choice for the teacher model. In previous deep learning works, network structure and parameters often need to be adjusted according to practical application [8,10]. It relies on users' experience and usually needs many experiments. If the whole training process can be properly designed, U-Net can achieve good results in most cases [12]. So it seems that the most straightforward way to build a student model is initializing another nn-UNet model with different initial parameters. However, training nnU-Net cost a lot of time, and its inference efficiency may not meet practical

demands, we do not use it as our final choice. Despite this, from the well-trained nnU-Net model, we can get strong pseudo labels of the unlabeled data. We use these pseudo labels and unlabeled data as a new training set to modulate a new student model built with the vision transformer. In this section, we first introduce nnU-Net briefly and then bring out our new student model.

### 2.1   nnU-Net

Isensee et al. proposed nnU-Net [7], which can adapt to many datasets in a supervised training process. nnU-Net adjusts the network structure according to the characteristics of the training set. It can process images with various shapes and textures, so as to achieve SOTA results in multiple medical segmentation tasks [1]. Specifically, for different datasets, nnU-net defines adaptive adjustment strategies from four perspectives, including preprocessing, training procedure, inference, and postprocessing.

**Network structure** nnU-Net consists of 2D U-Net, 3D U-Net, and U-Net Cascade. In these architectures, ReLU is replaced with Leaky ReLU and batch normalization is replaced with instance normalization. while the network structure remains almost the same as the default U-Net and it did not adopt additional modules such as attention mechanisms.

3D U-Net is usually used for training on 3D medical images, including CT and MRI. However, it occupies a large amount of GPU memory. In order to improve training speed and reduce resource consumption, the patch-based 3D U-Net can be adopted to reduce the cost of network computing. 3D U-Net is mainly to solve the problem of the poor effect of 2D U-Net in anisotropic data. On the other hand, the patch-based 3D U-Net may have a poor effect on large image sizes due to a limited global view. 3D U-Net Cascade is used to solve this problem.

The network topologies adjust adaptively according to the image size. It considers the image geometry and balances the GPU memory occupation which corresponds to the adjustment of the network capacity and batch size. The initial network configuration is as follows:

2D U-Net: An input patch size is set to $256 \times 256$, a batch size of 42, and the number of feature maps of the highest layer is set to 30 (the number of feature maps will be doubled with each downsampling). The network parameters are automatically adjusted to the median plane size of each dataset so that the network can effectively train the whole slice.

3D U-Net: An input patch size is set to $128 \times 128 \times 128$, a batch size of 2, and the number of feature maps at the highest level is 30. Due to the GPU memory limitation, the resolution of the image size beyond $128^3$ voxels is not increased but matches the median voxel size of the input image. If the median shape of the dataset is smaller than $128^3$, we use the median shape as the input image size and add batch size.

U-Net Cascade: The first level 3D U-Net is firstly trained on the down-sampled image and then the results are up-sampled to the original resolution. These results are fed into the second level 3D U-Net.

**Preprocessing** Image preprocessing is a very important part of training. For nnU-Net teacher models, this process is divided into three steps: (1) Cropping: Crop all data to the non-zero area. (2) Resampling: In order to enable the network to learn spatial semantics, images are resampled to the median voxel spacing of the dataset, and third order spline interpolation and nearest neighbor interpolation methods are used for data and segmentation mask respectively. (3) Normalization: For CT images, pixel values within the segmentation mask are collected, and all data is truncated to [0.5, 99.5] percentiles of these pixel values, followed by a z-score normalization. If the average size is decreased by more than 1/4, normalization is only applied to non-zero elements of the mask, and values outside the mask are set to 0.

### 2.2 Semi-supervised Cascaded Organ Detection, Identification and Segmentation

The overall architecture of our inference pipeline, i.e., the student part in the whole semi-supervised framework, is shown in Fig. 1. It consists of three stages. First, we adopt a lightweight U-Net to obtain the abdomen region-of-interest (RoI). Then we locate each organ with a new detection network built upon a vision transformer. Finally, we segment organs according to the detection bounding boxes. In the following context, we first describe the pseudo label preparation process, then we will introduce our inference architecture stage-by-stage according to Fig. 1.

**Pseudo Label preparation** The quality of pseudo labels is the key to determining whether the use of unlabeled data in semi-supervised training is effective. Poor quality pseudo labels may mislead the student model to learn wrong semantic information. In order to acquire high-quality pseudo labels, we adopt a consistency voting strategy that measures the consistency between pseudo labels generated by different teachers for the same case. The insight in our strategy is straightforward. For example, simple cases should be easy for most teacher models whereas hard cases may cause most models to fail. If a case causes different models to produce very inconsistent prediction outputs, we think that the distribution of this example is likely to be outside the distribution of most examples. We, therefore, reject examples with inconsistent pseudo labels, as they are likely to mislead the student network.

We choose nnU-Net as the teacher model. In order to enhance consistency between different teacher models, We build multiple different nnU-Net models with different initialization parameters. The same architecture of these teacher models ensures better consistency. Then we train these teacher models using 50 labeled data, and the models obtained are not used for the final testing

stage, but only for generating pseudo labels. The mean DSC of the results on the validation set exceeds 0.89. We believe that the nnU-Net models have been able to generate pseudo labels of good quality. We take 2000 unlabeled data as input and use trained nnU-Net models to generate corresponding pseudo labels. Finally, we measure the consistency between these pseudo labels and screen high-quality pseudo labels. After a segmentation results ensemble, we obtain our framework's final pseudo labels as input labels.

**Abdomen RoI Extraction** Given labels and pseudo labels, we train a simplified U-Net model to identify organ regions, then the proper RoI can be inferred by calculating the weighted average coordinates and distribution scope of the predicted organ voxels. This step helps us filter irrelevant background regions.

**Organ Detection and Identification** We propose a new detection framework based on DETR [2] to detect each organ. DETR handles object detection as a direct set prediction problem through the conjunction of the bipartite matching loss and transformer with parallel decoding of queries. In abdomen CTs, the number and relative position of organs are stable. We intend to estimate a bounding box for each organ to obtain an accurate and compact RoI. To this end, we estimate bounding boxes, based on the comprehensively annotated instance-level organ segmentation mask (ground-truth label and pseudo label), as supervision signal. For queries matched to the background class, only classification loss is accounted for.

**Organ Segmentation** To get high accuracy instance segmentation results, we adopt multiple stand-alone U-Net [12] models to segment each organ independently with a finer spatial resolution but in a locally cropped patch based on the detected bounding boxes from the second stage. The segmentation heads perform a binary segmentation for all 3D patches. After this, all predicted binary masks are merged back with their corresponding labels and spatial locations to form the final instance segmentation results of organs.

**Preprocessing** Before training the student model, we conduct preprocessing similar to the preprocessing used for the teacher model (nnU-Net). We perform cropping, resampling and normalization.

**Training Procedure** The model is trained from scratch and evaluated by five-fold cross-validation on the training set. The total loss for segmentation is Dice loss [4] combined with cross-entropy.

$$\mathcal{L}_{total} = \mathcal{L}_{dice} + \mathcal{L}_{CE} \tag{1}$$

For the teacher model, Adam is selected as the optimizer in the training process, with an initial learning rate of $3 \times 10^{-4}$ and 250 batches of each epoch.

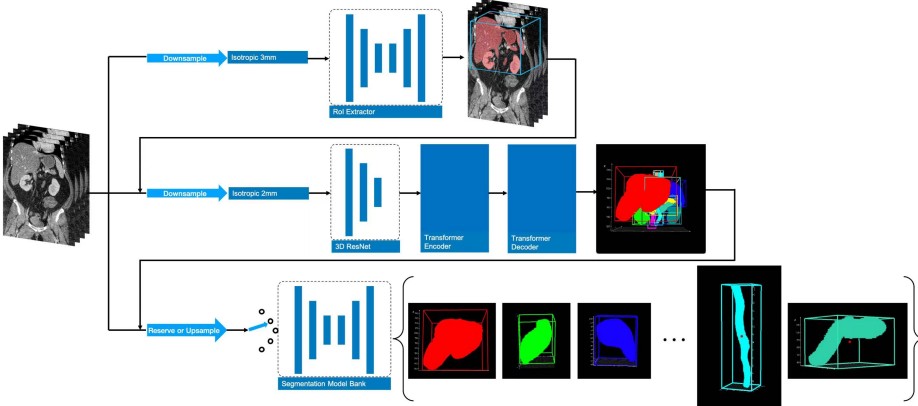

**Fig. 1.** Schematic of our inference architecture.

a Learning rate adjustment strategy is used which calculates the exponential moving average loss of the training set and validation set. If the training set loss decreases less than $5 \times 10^{-3}$ within 30 epochs, then the learning rate decreases by 5 times. When the learning rate is larger than $10^{-6}$ and the exponential moving average loss of the validation set decreases less than $5 \times 10^{-3}$ within 60 epochs, the training is terminated. Random rotations, Random scaling, Random elastic deformations, Gamma correction augmentation, and Mirroring are adopted as data augmentation. If the maximum side length of the image patch size of 3D U-Net is more than twice the minimum side length, then 2D data augmentation methods are used. For the student model, readers are referred to 2 for stage-specific training details.

**Inference** All inferences are performed by the student model. In our implementation, we dynamically clear the memory footprint to release the redundant memory occupancy in time and reduce resource consumption. The inference speed of our method is very fast thanks to the cascaded detection-then-segmentation strategy which significantly reduces the computation cost of redundancy regions.

**Postprocessing** For the teacher model, we adopt commonly used postprocessing methods such as removing small connected components. It is generally considered that a certain class is within a simply connected domain, which means that there is only one such domain within a case. So only the largest connected domain is retained, and the other small connected domains are removed. For the student model, we omit the postprocessing step for the sake of inference efficiency.

## 3    Experiments

### 3.1    Dataset and evaluation measures

The FLARE 2022 dataset is curated from more than 20 medical groups under the license permission, including KiTS [6] and AbdomenCT-1K [9]. The training set includes 50 labeled CT scans with pancreas disease and 2000 unlabeled CT scans with liver, kidney, spleen, or pancreas diseases. The validation set includes 50 CT scans with liver, kidney, spleen, or pancreas diseases. The testing set includes 200 CT scans where 100 cases have liver, kidney, spleen, or pancreas diseases and the other 100 cases have uterine corpus endometrial, urothelial bladder, stomach, sarcomas, or ovarian diseases. All the CT scans only have image information and the center information is not available.

The evaluation measures consist of two accuracy measures: Dice Similarity Coefficient (DSC) and Normalized Surface Dice (NSD), and three running efficiency measures: running time, area under GPU memory-time curve, and area under CPU utilization-time curve. All measures will be used to compute the ranking. Moreover, the GPU memory consumption has a 2 GB tolerance.

### 3.2    Implementation details

**Environment settings** We develop our cascaded model based on PyTorch [11]. All models are trained from scratch. We train the segmentation networks with a combination of dice and cross-entropy loss. We use the AdamW optimizer in the detection part and the Adam optimizer in the RoI extractor and segmentation part. An initial learning rate of $1 \times 10^{-4}$ is used in RoI extractor, $4 \times 10^{-4}$ and $1 \times 10^{-3}$ are used respectively in detection and segmentation. Training batches are set as 8, 8, and 4 respectively. The development environments and requirements are presented in Table 1.

**Table 1.** Development environments and requirements.

| | |
|---|---|
| Windows/Linux version | AliOS 7 |
| CPU | Intel(R) Xeon(R) Platinum 8163 CPU @ 2.50GHz |
| RAM | 724GB |
| GPU (number and type) | Eight Tesla V100 32G |
| CUDA version | 11.4 |
| Programming language | Python 3.7.3 |
| Deep learning framework | PyTorch (torch 1.7.0, torchvision 0.8.1) |

**Training protocols** All images are automatically normalized based on statistics of the entire respective dataset. During training, in order to help networks properly learn spatial semantics, all patients are resampled to the median voxel spacing of their respective dataset, where third-order spline interpolation is used

for image data and nearest-neighbor interpolation for the corresponding segmentation mask. The detailed training protocols are shown in Table 2.

**Table 2.** Training protocols. "roi" means the RoI extraction in stage 1. "det" means the organ detection network in stage 2. "seg" means the segmentation head in stage 3.

| Network initialization | Kaiming normal initialization |
|---|---|
| Batch size | roi: 8 \| det: 8 \| seg: 4 |
| Patch size | seg only: organ-specific patch size |
| Total epochs | roi: 1000 \| det: 1000 \| seg: 500 |
| Optimizer | roi: Adam \| det: AdamW \| seg: Adam |
| Initial learning rate (lr) | roi: 0.0001 \| det: 0.0004 \| seg: 0.001 |
| Lr decay schedule | warmup 200 epochs and ×0.1 at 800th epoch |
| Training time | roi: 52h \| det: 20h \| seg: organ-specific |
| Number of model parameters | roi: 4.8M \| det: 9.6M \| seg: 4.8M |
| Loss function | seg: Dice loss and cross entropy |

## 4  Results and discussion

### 4.1  Quantitative results on validation set

We compare our method with two state-of-the-art segmentation models including CNN-based methods and vision transformer-based methods. As shown in Table 3, our results currently obtain 0.897 mean DSC on the leaderboard of FLARE 2022 validation set. Compared with nnU-Net and Swin-UNETR [13], which are also trained from scratch, our method exceeds these two methods in terms of DSC on most abdominal organs. Moreover, our model is even better than the Swin-UNETR model with pre-training on FLARE unlabeled part. This emphasizes the significance of our semi-supervised method. Last but not least, our method even outperforms the ensembled nnU-Net, which ensemble the segmentation results of 12 different initialized nnU-Net models, and has much less training and inference time than the nnU-Net with the ensemble. The segmentation results of these methods are shown in Fig. 2. Our method can obtain better segmentation results than all the other methods. For hidden testing set, our method obtains 0.889 mean DSC and 0.933 mean NSD as shown in Table 4.

### 4.2  DSC comparisons between with and without unlabeled images

Due to the long training and inference time of nnU-Net, we only use it to generate pseudo labels of 2000 unlabeled images. Then we use these unlabeled images and their pseudo labels to train our model. In order to validate the effectiveness of the unlabeled images and the pseudo labels, we conduct an ablation study on

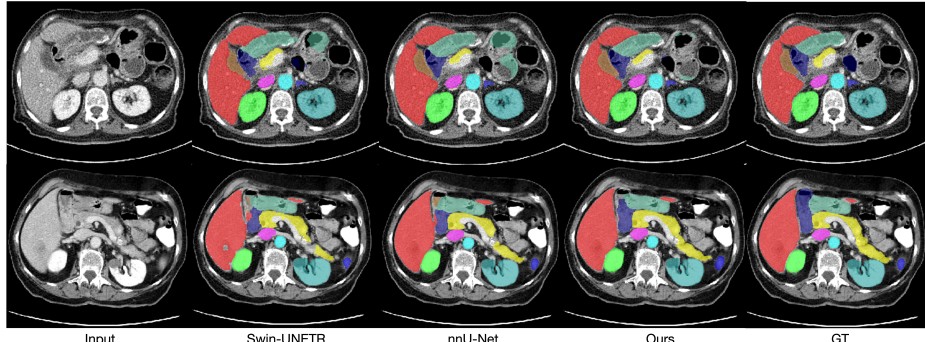

Input          Swin-UNETR          nnU-Net          Ours          GT

**Fig. 2.** Comparison between segmentation results of different methods.

**Table 3.** DSC values on different organs. Abbreviations: "Liv."-Liver, "RK"-Right Kidney, "Spl."-Spleen, "Pan."-Pancreas, "Aor."-Aorta, "IVC"-Inferior Vena Cava, "RAG"-Right Adrenal Gland, "LAG"-Left Adrenal Gland, "Gall."-Gallbladder, "Eso."-Esophagus, "Sto."-Stomach, "Duo."-Duodenum, "LK"-Left Kidney.

| Methods | Liv. | RK | Spl. | Pan. | Aor. | IVC | RAG | LAG | Gall. | Eso. | Sto. | Duo. | LK | mDSC |
|---|---|---|---|---|---|---|---|---|---|---|---|---|---|---|
| Swin-UNETR | 0.965 | 0.912 | 0.942 | 0.846 | 0.930 | 0.865 | 0.758 | 0.742 | 0.771 | 0.790 | 0.886 | 0.765 | 0.887 | 0.850 |
| Swin-UNETR pre. | 0.964 | 0.921 | 0.952 | 0.881 | 0.937 | 0.862 | 0.794 | 0.791 | 0.792 | 0.818 | 0.895 | 0.790 | 0.879 | 0.867 |
| nnU-Net | 0.977 | 0.941 | 0.958 | 0.872 | 0.968 | 0.878 | 0.830 | 0.801 | 0.765 | 0.892 | 0.899 | 0.771 | 0.911 | 0.882 |
| nnU-Net ens. | 0.979 | **0.948** | 0.960 | 0.886 | **0.969** | 0.897 | **0.838** | **0.819** | 0.787 | **0.901** | 0.907 | **0.792** | 0.920 | 0.892 |
| Ours | **0.980** | 0.945 | **0.972** | **0.890** | 0.966 | **0.903** | 0.824 | 0.806 | **0.861** | 0.874 | **0.915** | 0.787 | **0.937** | **0.897** |

our organ detection module, which is relatively more sensitive to the amount of data due to its task attribute and transformer component. As shown in Table 5, the effect of using unlabeled cases is significant. If we remove the training of unlabeled images with pseudo labels, we observe a significant performance drop in our final results. Since there are few labeled images, the distribution of labeled images is very different from the real data distribution. So if we do not use the unlabeled images, the model will have no chance to learn unseen cases in the target data distribution. This adds huge difficulties to regress 3D boxes and segment accurate boundaries for organs, especially for relatively small organs such as gallbladder and adrenal glands.

### 4.3   Visualized examples of successful and failed cases

Fig. 3 shows the segmentation results of our method. It clearly reveals that our method can obtain excellent segmentation results on most organs. However, we find that sometimes the model failed especially when some organs have larger size and shape variations due to the appearance of tumors. For example, the trained models can't generalize well when the patient has a kidney tumor, which makes the size of the kidney much larger than usual. One possible solution is adding more supervised cases which have a similar distribution to those hard cases.

**Table 4.** Final results on the hidden test set.

| Methods | DSC(%) | NSD(%) | Time(s) | GPU(MB) | CPU(%) |
|---------|--------|--------|---------|---------|--------|
| Ours    | 0.889  | 0.933  | 27.32   | 6028    | 533.1  |

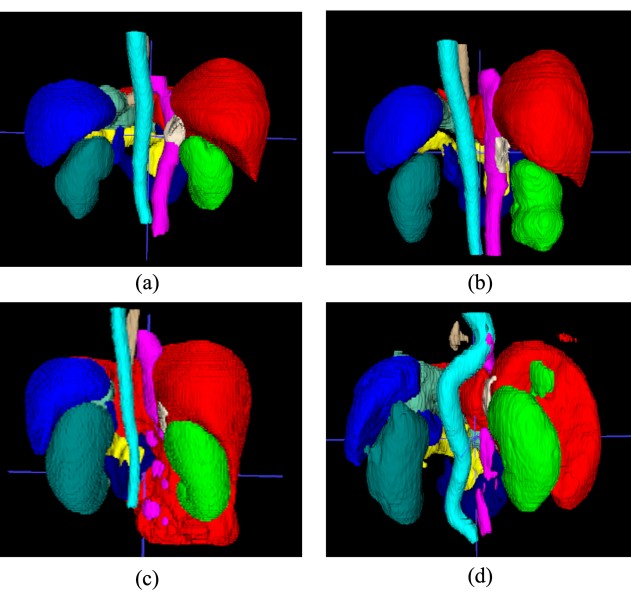

(a)                    (b)

(c)                    (d)

**Fig. 3.** (a) to (b): Plots of good results visualization and (c) to (d): Plots of bad results visualization.

### 4.4   Segmentation efficiency analysis

We perform segmentation efficiency analysis on validation set, the results are shown in Table 6. Our method is significantly faster than other methods in terms of inference time. To be noted, our inference time measurement does not include Docker launching stage and model initialization, because there may exist large variance among different configurations. Besides, we start the Docker only once at the start of the evaluation, and get the average inference time of all evaluation cases. Therefore, the measurement of our method in Table 6 is smaller than that in Table 4.

### 4.5   Limitations and future work

The proposed method works well on most cases. However, there are still some misclassification failures on some organs. Perhaps adding organ shape-related prior knowledge will help solve the limitations, which is left for future work.

**Table 5.** DSC comparisons between with and without using unlabeled images. *wo.* means without using unlabeled images and *w.* means using unlabeled images.

| Methods | Liv. | RK | Spl. | Pan. | Aor. | IVC | RAG | LAG | Gall. | Eso. | Sto. | Duo. | LK | mDSC |
|---|---|---|---|---|---|---|---|---|---|---|---|---|---|---|
| Ours *wo.* | 0.975 | 0.885 | 0.879 | 0.876 | 0.952 | 0.898 | 0.809 | 0.753 | 0.594 | 0.855 | 0.862 | 0.770 | 0.881 | 0.845 |
| Ours *w.* | 0.980 | 0.945 | 0.972 | 0.890 | 0.966 | 0.903 | 0.824 | 0.806 | 0.861 | 0.874 | 0.915 | 0.787 | 0.937 | 0.897 |

**Table 6.** Efficiency analysis of different methods.

| Methods | Inference time (s) | GPU memory footprint (MB) |
|---|---|---|
| Swin-UNETR | 18.00 | 22284 |
| nnU-Net | 126.40 | 4639 |
| Ours | 3.10 | 3208 |

## 5  Conclusion

In this paper, we propose a novel three-stage instance segmentation network for the abdominal organ segmentation task. We develop and test the whole framework on the FLARE 2022 challenge dataset. The network consists of a vision transformer-based detection model and several lightweight segmentation heads. We adopt a semi-supervised learning strategy to leverage a large amount of unlabeled data. We use nnU-Net as the teacher model and design a consistency measuring strategy to generate high-quality pseudo labels. The whole framework of our method acquires 0.897 mean DSC on the FLARE 2022 challenge validation dataset.

**Acknowledgements** The authors of this paper declare that the segmentation method they implemented for participation in the FLARE 2022 challenge has not used any pre-trained models nor additional datasets other than those provided by the organizers. The proposed solution is fully automatic without any manual intervention.

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
