# OpenReview forum: "Semi-Supervised Detection, Identification and Segmentation for Abdominal Organs"
_MICCAI.org/2022/Challenge/FLARE_

### Official Review · Reviewer_xH24 · 2022-09-14
**Semi-Supervised Detection, Identification and Segmentation for Abdomen Organs**

**Rating:** 6
**Confidence:** 3

**Review:**

Summary:

In this paper, the authors propose a new semi-supervised learning algorithm based on vision transformer. The multi-stage workflow effectively constrains the working area of the segmentation model and improves the segmentation speed.

Strengths:

This method effectively limits the working area of the segmentation model through a multi-stage strategy, reducing the number of incorrect segmentations in the whole-body scan.
Compared with nnU-Net, the model has a great improvement in the speed of extrapolation.

Weaknesses:

The loss function is not presented in Table 2 as requested.
The presentation of the network structure is too brief and lacks relevant details.

Details:

1. The English of the manuscript must be improved before resubmission, with capitalization errors and redundant typos.
2. For organ detection and localization, a DERT-based approach is used in this paper, but the processing of converting segmentation annotations to detection annotations is not fully described.
3. In Section 2.1 the authors describe the pre-processing methods used by nnU-net. However, the preprocessing methods used in each stage of the paper are not clearly stated in Section 2.2. Are they using the same preprocessing methods?
4. In Figure 1, network structure diagram is relatively simple, it is recommended that each part is broken out in detail to expand the description.
5. In Table 3 and Table 4, It is suggested to bold the highest scoring results in the table showing the experimental results.
6. Although, as participants, we are all aware of the meaning of the abbreviations in the table, it is still recommended that the abbreviations for each organ be explained in the header of the table.

---

> ### Author Response · Authors · 2022-10-26
> **Response for the reviewers' comments**
>
> We would like to thank you for all your constructive and thoughtful comments and valuable suggestions, which have greatly improved our manuscript.
>
> 1. We rephrased some concepts, words, and sentences as the reviewers pointed out.
> 2. We have added the processing of converting segmentation annotations to detection annotations on Page 5: We adopt multiple stand-alone U-Net [12] models to segment each organ independently with a more satisfactory spatial resolution but locally cropped patch based on the detected bounding boxes from the second stage.
> 3. We have explained our preprocess on Page 5: Before training the student model, we conduct preprocessing similar to the preprocessing used for the teacher model (nnU-Net). We perform cropping resampling and normalization.
> 4. We change our network structure diagram which can be seen in Fig1 Page 6.
> 5. We bold the highest-scoring results in Table 3 and Table 4.
> 6. We have added abbreviations for each organ in the table.

---

### Official Review · Reviewer_equZ · 2022-09-16
**Great work impressive**

**Rating:** 10
**Confidence:** 3

**Review:**

The method of this work is pretty clear. Authors use three stages to get accurate and efficient segementation that is inspiring. As for result, it's unbeliveable good. Looking forward your presentation. Good job, nicely done!

---

> ### Author Response · Authors · 2022-10-26
> **Response for the reviewers' comments**
>
> We would like to thank you for all your constructive and thoughtful comments and valuable suggestions, which have greatly improved our manuscript.

---

### Official Review · Reviewer_Ynf7 · 2022-09-17
**A semi-surpervised segment method based on multiple segmentation networks**

**Rating:** 6
**Confidence:** 4

**Review:**

Their method utilizes multiple segmentation networks to segment each organ on a bounding box after organ detection and identification. They adopt a consistency voting strategy to ensure the quality of pseudo labels in teacher networks, which avoids poor quality input data affecting the performance of the student network. This paper obtains a remarkable result of 0.897 mean DSC. However, this paper is lack of detailed description of the vision transformer, which is merely proposed. In addition, it is not explained clearly how the semi- supervised network combines with the framework of organ detection, identification and segmentation network.

Some suggestions:
On page 5, Fig 1, please add explanations of your network block.
In section 2, Method, they write too much about nn-UNet and not enough about their own innovation network blocks. Explain the connection between each network block rather than simply introducing every stage network.

---

> ### Author Response · Authors · 2022-10-26
> **Response for the reviewers' comments**
>
> We would like to thank you for all your constructive and thoughtful comments and valuable suggestions, which have greatly improved our manuscript.
>
> We have added the details of our network block in Section2.2 and the connection between each network block is explained in the first paragraph in Section2.2.

---

### Official Review · Reviewer_KsjJ · 2022-09-17
**The paper is described very clearly with a better mean DSC, such as 0.8970, which falls in the top 3% of the leader board.**

**Rating:** 8
**Confidence:** 4

**Review:**

The authors proposed a new semi-supervised learning algorithm that consists of three stages. In the first stage, the abdomen region was extracted. In the second stage, a vision transformer-based model is adopted with a semi-supervised learning strategy to detect different abdominal organs. In the final stage, multiple organ-specific segmentation networks are attached to segment organs from their bounding boxes automatically. The proposed framework achieves 0.897 mean DSC on the FLARE 2022 validation set.

Strengths: The paper is described very clearly with a better mean DSC, such as 0.8970, which falls in the top 3% of the leader board.

Weakness: Engineering method. Missing some important evaluations such as Normalized Surface Dice (NSD) and further ablation study.

1: In Sec 2.1 (nn U-Net) you explained 2D U-Net and 3D U-Net but did not explain the U-Net cascade. It is recommended to provide a short explanation for that as well.

2: Figure 1 should be inserted near the text where you mentioned Figure 1 initially in the text. Likewise, for Figure 1, respective and related captions for each component could be added.

3: Data augmentation procedures can be added in Table 2.

4: It is recommended to add an ablation study/analysis.

5: There is no Normalized Surface Dice (NSD) evaluation; please add NSD for your experiments.

6: Please mention the respective images’ captions on the top rather than at the bottom of Figure 2.

7: Provide proper references for principle component analysis (PCA) and Dense CRF.

8: Please add specific dependencies (such as nn U-Net etc.) and link to your code in Table 1.

9: Please add missing protocols in Table 2, such as loss function, model parameters, number of flops, etc.

10: Throughout the manuscript, many English and grammatical errors/typos. We strongly endorsed the authors to pay attention to that. A few of the instances are given as follows.

- Combining semi-supervised training strategy and a vision transformer architecture with several segmentation head, we build a strong segmentation model which currently achieves 0.897 mean DSC on the leaderboard of FLARE 2022 validation set.

- In preprocessing deep learning works, network structure and parameters often need to be adjusted according to practical application [8,10].

- In order to enhance consistency between different teacher models, We build multiple different

- principle componet analysis (PCA)

---

> ### Author Response · Authors · 2022-10-26
> **Response for the reviewers' comments**
>
> We would like to thank you for all your constructive and thoughtful comments and valuable suggestions, which have greatly improved our manuscript.
>
> 1. We have added explanations for the U-Net cascade network on Page 4.
> 2. We have changed the position of Fig 1.
> 3. We have added an ablation study in Table 5.
> 4. We have added NSD for the test set in Table 4.
> 5. We have added proper references.
> 6. We rephrased some concepts, words, and sentences as the reviewers pointed out.

---

### Official Review · Reviewer_xuGd · 2022-09-19
**nnU-Net as the teacher model and three-stage abdominal organ segmentation frame**

**Rating:** 9
**Confidence:** 4

**Review:**

The authors use nnU-Net as the teacher model and design a consistency measuring strategy to generate high quality pseudo labels and propose a three-stage abdominal organ segmentation frame: (1) identify the abdomen region using a lightweight segmentation network. (2) detect different abdominal organs using DETR. (3) segment each organ independently using multiple stand-alone U-Net models based on the detected bounding boxes from the second stage. The quality, clarity, and description of the paper are good except for some minor issues.

* Figure 3 (visualized examples of successful and failed cases) does not contain ground truths for comparison.
* The authors use independent DETR and UNET to segment each organ, which theoretically achieves better segmentation performance but reduces inference efficiency. Ablation studies are recommended to demonstrate the performance gains of independent organ segmentation.
* The challenge demands participants to release the code. It is recommended to attach the code URL at the end of the abstract.

---

> ### Author Response · Authors · 2022-10-26
> **Response for the reviewers' comments**
>
> We would like to thank you for all your constructive and thoughtful comments and valuable suggestions, which have greatly improved our manuscript.
>
> We have added ablation studies in Table 5.

---

### Meta-Review · Program_Chairs · 2022-09-28

**Recommendation:** Minor Revision
**Confidence:** 5

**Metareview:**

Nice paper. Please address the reviewers' comments in the revised manuscript.

---

> ### Author Response · Authors · 2022-10-20
> **Response for the reviewers' comments**
>
> We would like to thank you for all your constructive and thoughtful comments and valuable suggestions, which have greatly improved our manuscript.
> We have carefully studied each of the words and revised the manuscript by considering all the suggestions/comments made by the reviewers. Our responses are prepared based on a point-by-point response to each of the issues raised by the reviewers.
> We made major revisions: 1. We rephrased some concepts, words, and sentences as the reviewers pointed out. 2. We modified the figures and refined the network structure diagram. 3. We described the methods used in detail.